# Thiolated Chitosan Conjugated Liposomes for Oral Delivery of Selenium Nanoparticles

**DOI:** 10.3390/pharmaceutics14040803

**Published:** 2022-04-06

**Authors:** Atiđa Selmani, Elisabeth Seibert, Carolin Tetyczka, Doris Kuehnelt, Ivan Vidakovic, Karin Kornmueller, Markus Absenger-Novak, Borna Radatović, Ivana Vinković Vrček, Gerd Leitinger, Eleonore Fröhlich, Andreas Bernkop-Schnürch, Eva Roblegg, Ruth Prassl

**Affiliations:** 1Department of Pharmaceutical Technology and Biopharmacy, Institute of Pharmaceutical Sciences, University of Graz, 8010 Graz, Austria; atida.selmani@uni-graz.at (A.S.); carolin.tetyczka@uni-graz.at (C.T.); eva.roblegg@uni-graz.at (E.R.); 2Division of Biophysics, Gottfried Schatz Research Center for Cell Signaling, Metabolism and Aging, Medical University of Graz, 8010 Graz, Austria; elisabeth.seibert@uni-graz.at (E.S.); ivan.vidakovic@medunigraz.at (I.V.); karin.kornmueller@medunigraz.at (K.K.); 3Institute of Chemistry, Analytical Chemistry, NAWI Graz, University of Graz, 8010 Graz, Austria; doris.kuehnelt@uni-graz.at; 4Center for Medical Research, Medical University of Graz, 8010 Graz, Austria; markus.absenger@medunigraz.at (M.A.-N.); eleonore.froehlich@klinikum-graz.at (E.F.); 5Center of Excellence for Advanced Materials and Sensing Devices, Institute of Physics, 10000 Zagreb, Croatia; bradatovic@ifs.hr; 6Institute for Medical Research and Occupational Health, 10000 Zagreb, Croatia; ivinkovic@imi.hr; 7Division of Cell Biology, Histology and Embryology, Gottfried Schatz Research Center for Cell Signaling, Metabolism and Aging, Medical University of Graz, 8010 Graz, Austria; gerd.leitinger@medunigraz.at; 8Department of Pharmaceutical Technology, Center for Chemistry and Biomedicine, Institute of Pharmacy, University of Innsbruck, 6020 Innsbruck, Austria; andreas.bernkop@uibk.ac.at

**Keywords:** thiomers, microfluidic synthesis, biophysical characterization, in vitro intestinal model

## Abstract

This study aimed to design a hybrid oral liposomal delivery system for selenium nanoparticles (Lip-SeNPs) to improve the bioavailability of selenium. Thiolated chitosan, a multifunctional polymer with mucoadhesive properties, was used for surface functionalization of Lip-SeNPs. Selenium nanoparticle (SeNP)-loaded liposomes were manufactured by a single step microfluidics-assisted chemical reduction and assembling process. Subsequently, chitosan-N-acetylcysteine was covalently conjugated to the preformed Lip-SeNPs. The Lip-SeNPs were characterized in terms of composition, morphology, size, zeta potential, lipid organization, loading efficiency and radical scavenging activity. A co-culture system (Caco-2:HT29-MTX) that integrates mucus secreting and enterocyte-like cell types was used as a model of the human intestinal epithelium to determine adsorption, mucus penetration, release and transport properties of Lip-SeNPs in vitro. Thiolated Lip-SeNPs were positively charged with an average size of about 250 nm. Thiolated Lip-SeNPs tightly adhered to the mucus layer without penetrating the enterocytes. This finding was consistent with ex vivo adsorption studies using freshly excised porcine small intestinal tissues. Due to the improved mucoadhesion and retention in a simulated microenvironment of the small intestine, thiolated Lip-SeNPs might be a promising tool for oral selenium delivery.

## 1. Introduction

Selenium (Se) is an essential micronutrient playing important roles in many physiological processes and metabolic pathways [1,2]. As a key component of selenoproteins, Se has a protective role in the etiology of various diseases including cancer, Alzheimer’s and Parkinson disease, diabetes, cardiovascular and intestinal diseases [3,4,5,6]. Numerous studies demonstrated beneficial effects of Se in DNA repair, endocrine system function and immune response, as well as pronounced anti-cancer, anti-microbial, anti-inflammatory and antioxidant properties [7,8,9]. The subtle balance between Se deficiency and toxicity led to controversial results for nutritional Se supplementation with evidence of potential adverse effects related to excess Se intake [4]. In its inorganic forms, Se has a narrow therapeutic window between healthy and toxic dosages, while elemental Se has a lower toxicity and improved bioavailability, but also very poor solubility [10,11]. Nanotechnology based design may open up the possibility to fabricate materials with improved and tunable physicochemical properties compared to bulk materials that are formulated at the microscale [12,13]. Accordingly, Se nanoparticles (SeNPs) have been recognized as innovative nanomaterials with enhanced biological efficacy compared to other bulk Se forms [14]. They are discussed as promising entities for the design and development of novel medical products and nanomedicines [15,16]. Different agents such as polymers, surfactants, proteins or polysaccharides have been reported for surface stabilization of SeNPs in order to advance colloidal stability, to improve bioavailability and to reduce toxicity [17,18,19,20,21]. In particular, biopolymers such as amphiphilic multiblock copolymers or modified poly-ethylene glycols have been shown to be highly successful in stabilizing metallic nanoparticles [22,23].

For oral administration, the surface coating might be adjusted to prevent particle degradation in the harsh conditions of the gastrointestinal (GI) microenvironment and to enhance adhesion and penetration of drugs through GI-barriers. Thiolated polymers, designated thiomers, are a class of polymers that are comprised of immobilizing agents with thiol groups coupled to polymeric backbones [24,25]. They have improved mucoadhesive properties and belong to the most promising coating agents in the design and formulation of oral nano-drug delivery systems [26,27,28,29]. Thiomers can also be used to coat liposomal surfaces with polymers [30]. Likewise, Gradauer et al., developed chitosan-thioglycolic acid coated liposomes with enhanced mucoadhesive, permeation enhancing and efflux pump inhibitory properties when compared to uncoated liposomes. The thiolated liposomes were non-toxic, stable in simulated gastrointestinal fluids and showed sustainable release of drugs in the small intestine as well as a non-immunogenic response in rats [31,32,33]. All these data indicate that thiomer coated liposomes may meet the required criteria for an efficient delivery system for SeNPs. So far, only a few studies reported the use of liposomes for SeNPs encapsulation. Youngren et al. [34] developed a formulation of mannosylated Se loaded liposomes (Man-Lip-Se) for targeting the immune system. The Man-Lip-Se were stable in simulated gastric and intestinal fluids and exhibited controlled release of Se. Ahmed et al. [35] used multilamellar SeNPs loaded liposomes in a pre-clinical study to determine the potential of SeNPs to treat diabetes mellitus. Xie et al. [36] designed Se-functionalized liposomes loaded with doxorubicin for cancer treatment. Their particles exhibited continuous release of doxorubicin with increased cellular uptake, while synergistic behavior of the anticancer drug and SeNPs enhanced the overall anticancer effect of the formulation. This work represents a compelling example of how a smart combination of nanoparticles and drugs encapsulated into the same liposomal particles can effectively improve the therapeutic outcome.

The aim of this study was to design a hybrid thiolated chitosan coated liposomal delivery system for SeNPs to enhance the bioavailability of Se in the microenvironment of the small intestine. SeNPs were synthesized and loaded into liposomes in a single-step manufacturing process using a microfluidics based chemical reduction and assembling procedure. After covalent coupling of chitosan-N-acetylcysteine (Cs-NAC), the Lip-SeNPs were characterized by means of size, surface charge and morphology, and evaluated in terms of loading capacity, cytotoxicity and radical scavenging behavior. Adsorption and release properties were determined under in vitro settings using a co-culture cell model of the small intestine, as well as under ex vivo settings using porcine intestinal tissues.

## 2. Materials and Methods

### 2.1. Materials

Palmitoyl-oleoyl-phosphatidylcholine (POPC), 1,2-dioleoyl-sn-glycero-3-phosphoethanolamine-N-[4-(p-maleimidomethyl)-cyclohexane-carboxamide] (DOPE-MCC) and 1,2,-dioleoyl-sn-glycero-3-phosphoethanolamine-N-(lissamine rhodamine B sulfonyl) were purchased from Avanti^®^ Polar Lipids (Alabaster, AL, USA). Chitosan-N-acetyl-cysteine (Cs-NAC) was synthesized and provided by Thiomatrix^®^ (Innsbruck, Austria). Sodium selenite (Na_2_SeO_3_) was purchased from Alfa Aesar^®^ (ThermoFisher GmbH, Kandel, Germany).

For Se analytics, analytical grade chemicals and ultrapure water (18.2 MΩ·cm, Academic water purification system, Millipore GmbH, Vienna, Austria) were used for the preparation of solutions. Nitric acid (65%, p.a.) was purchased from Carl Roth GmbH + Co. KG (Karlsruhe, Germany) or Chem Lab NV (Zedelgem, Belgium) and further purified by sub-boiling distillation. Single element standard solutions of selenium (1000 mg Se/L ± 0.2% in 2% HNO_3_), germanium (1000 mg Ge/L in 2% HNO_3_ and 0.5% HF) and indium (1000 mg In/L ± 0.2% in 2% HNO_3_) for total element determination by inductively coupled plasma mass spectrometry (ICP-MS) were obtained from Carl Roth GmbH + Co. KG. A human plasma sample from a round-robin exercise was used for quality control.

For cell culture experiments, a co-culture system consisting of two different cell lines was used. Caco-2 (ACC169, HTB-37 clone from the German Collection of Microorganisms and Cell Cultures) provided by E. Fröhlich (Medical University of Graz, Graz, Austria) and HT29-MTX-E12 cells kindly provided by T. Lesuffleur (INSERM UMR S 938, Paris, France) were used. Dulbecco’s Modified Eagle’s medium (DMEM), fetal bovine serum (FBS), penicillin streptomycin, and phosphate buffered saline (PBS; pH 7.4, ionic strength of 162.7 mM; 1.06 mM potassium phosphate monobasic, 155.17 mM sodium chloride and 2.97 mM sodium phosphate dibasic) were purchased from Gibco, Life Technologies Corporation (Painsley, Stoke-on-Trent, UK).

All other chemicals were of analytical grade and obtained from Sigma^®^ Aldrich (Vienna, Austria).

### 2.2. Methods

#### 2.2.1. Synthesis of SeNPs-Loaded Liposomes (Lip-SeNPs)

The assembling of the liposomes was performed via microfluidic rapid mixing as schematically shown in Figure 1 using the NanoAssemblr^®^ (Precision NanoSystems, Vancouver, BC, Canada). The lipid components were dissolved in 90% ethanol at a molar ratio of POPC:DOPE-MCC of 3:0.3. For in vitro and ex vivo studies, the fluorescent phospholipid 1,2-dioleoyl-sn-glycero-3-phosphoethanolamine-N-(lissamine rhodamine B sulfonyl) was added to the organic phase in a molar ratio of 3.3:0.01 (lipids to label). POPC liposomes served as control. The reduction of sodium selenite (Na_2_SeO_3_) to elementary selenium was performed with ascorbic acid, which was added to the organic phase in a molar ratio of ascorbic acid to selenium of 2.5:1. The aqueous phase consisted of a stock solution of 0.1 M Na_2_SeO_3_ and was diluted with double distilled 0.02 µm sterile filtrated water to reach a final Se concentration of 13 mM. For microfluidic based assembling of the liposomes, a flow ratio of 2.5:1 aqueous phase to organic phase and a flow rate of 5 mL/min were used. The final lipid concentration was about 3 mg/mL. To remove the ethanol, the samples were dialyzed (14 kD MWCO) for 24 h in PBS (10 mM, 150 mM NaCl, pH 7.4). For control experiments, empty liposomes without Se were prepared in the above described manner with double deionized water as aqueous phase.

#### 2.2.2. Coupling of Chitosan-N-Acetylcysteine (Cs-NAC)

The covalent coupling of Cs-NAC to the preformed liposomes was accomplished through the formation of thioether bonds between the free -SH groups of Cs-NAC to the maleimide groups of DOPE-MCC (Figure 2) [37]. The Cs-NAC conjugate was synthesized as described previously [38]. The amount of covalently attached thiol groups on chitosan was quantified photometrically with Ellman’s reagent (see Section 2.2.3) [39]. The modification of the chitosan was characterized by Fourier transform infrared spectroscopy (FTIR) using a Spectrum Two™ spectrometer (Perkin Elmer, Beaconsfield, United Kingdom). The displayed spectra are the mean of four scans measured from 4000 to 400 cm^−1^ at a resolution of 1 cm^−1^. Cs-NAC was further characterized by ^1^H NMR. The measurement was performed on a “Mars” 400 MHz Avance 4 Neo spectrometer from Bruker Corporation (Billerica, MA, USA, 400 MHz) in deuterium oxide (D2O) solution. The center of the broad solvent peak (δ = 4.75 ppm) was used as an internal standard. Cs-NAC was dissolved in PBS buffer at a concentration of 3 mg/mL and added to the liposomal suspension to obtain a final molar ratio of 1:3 (maleimide groups:SH-groups) [31]. The mixtures were incubated overnight at room temperature under shaking at 300 rpm (Multi Reax^®^, Heidolph Instruments GmbH & CO. KG, Schwabach, Germany).

#### 2.2.3. Determination of the Free -SH Groups via Ellman’s Assay

To determine the amounts of free -SH groups of Cs-NAC, a photometric assay—a slightly modified version of the Ellman’s test—was performed. Briefly, Ellman’s reagent 5,5′-dithiobis-2-nitrobenzoic acid (DTNB) was dissolved in water at a concentration of 7.5 mM and NaHCO_3_ was slowly added until the DTNB was fully dissolved. 2-mercaptoethanol was used as the standard reagent and standards in a concentration range from 25 µM to 5 mM was prepared. The assay was performed on a 96-well-plate (Cellstar^®^, Greiner Bio-One, Kremsmünster, Austria). Samples and standards were diluted to 160 µL with PBS buffer. Due to the orange color of SeNPs that interfered with the assay, non-selenium-loaded empty liposomes were used. An amount of 40 µL of the DTNB solutions were injected into each well of the plate. After an incubation time of 30 min in the dark, the absorbance was recorded at 412 nm with a Clariostar^®^ plate reader (BMG LABTECH GmbH, Ortenberg, Germany). To determine the amount of free thiol groups after coupling, the thiochitosan coupled samples with a molar ratio of 1:3 (maleimide groups:SH-groups) were compared to control POPC liposomes to which the same amount of Cs-NAC was added. Due to the lack of maleimide groups, Cs-NAC could not bind to the POPC liposomes and the initial amount of free -SH groups in Cs-NAC before coupling could be assessed.

#### 2.2.4. Particle Size and Zeta Potential Measurements

The particle size determination was carried out via Nanoparticle Tracking Analysis (NTA) using the Nanosight LM10 (Malvern Panalytical Ltd., Malvern, UK). During the measurements the temperature was kept constant at 25 °C and a green laser with a wavelength of 532 nm was used. The samples were diluted with double distilled, 0.02 µm filtrated water. For each measurement, five recordings of 60 s were taken and the particle size was analyzed from the recorded trajectories using the NTA software version 3.2. Particle diameters derived from the hydrodynamic radii are presented as mean particle diameter and as D50 and D90 values corresponding to 50% and 90% of the particle distribution, respectively, being below this size.

The zeta potential was determined using laser Doppler micro Electrophoresis (173° scattering angle) using a Zetasizer Nano ZS instrument (Malvern Panalytical Ltd., Malvern, UK) and calculated according to the Helmholtz–Smoluchovski equation. The measurements were performed at 25 °C in an aqueous sodium chloride solution (0.9% (*w*/*v*) NaCl, pH: 5.5–6.0, conductivity 50 mS/cm).

#### 2.2.5. Negative Staining Transmission Electron Microscopy (TEM)

The morphology of Lip-SeNPs was analyzed by TEM. The particles were dialyzed in HEPES buffer instead of PBS buffer to have a phosphate free suspension and diluted to 1:10 (*v*/*v*) with HEPES buffer. The samples were dropped onto a glow-discharged carbon-coated copper grid and allowed to settle down for 1 min. Excess fluid was carefully blotted with filter paper and immediately replaced by 10 µL of a 1% (*w*/*v*) uranyl acetate staining solution. After 1 min, the uranyl acetate solution was blotted and replaced by another 10 µL of fresh uranyl acetate solution, incubated for 1 min before the staining solution was removed, and the samples were air-dried. Visualization of the samples was performed with a FEI Tecnai G^2^ 20 transmission electron microscope (Eindhoven, The Netherlands) operating at an acceleration voltage of 120 kV. Digital images were acquired using a Gatan US1000 CCD camera at 2 k × 2 k resolution and the Digital Micrograph software (Version 1.93.1362, Gatan Inc., Pleasanton, CA, USA). Alternatively, a Zeiss EM900 TEM (Zeiss, Oberkochen, Germany) operating at 80 kV acceleration voltage was used. Images were taken with a CCD camera and processed with the ImageSP software (Version 1.2.8.64, TRS & Sys-prog, Minsk, Belarus) [40].

#### 2.2.6. Small-Angle X-ray Scattering (SAXS)

Synchrotron SAXS measurements were performed at the Austrian SAXS Beamline at ELETTRA, (Trieste, Italy) [41]. The scattering intensity was measured as a function of the scattering vector q, where q = 4π (sin θ)/λ, with 2 θ being the scattering angle and λ being the wavelength of 0.154 nm. An automatic sample injection system was used, with 15 µL of sample volume. Images were recorded with a Pilatus detector (Pilatus3 1 M, DECTRIS Ltd., Villigen PSI, Switzerland) calibrated with silver behenate. Uncoated and thiolated Lip-SeNPs with a lipid concentration of 3 mg/mL were measured in triplicates (12 images per sample with an exposure time of 10 sec each). All measurements were performed at a constant temperature of 20 °C. Data analysis was performed with Fit2D [42] and the program Igor Pro (Version 6.22A, WaveMetrics Inc., Portland, OR, USA). The scattering curves were fitted with a bilayer form factor model. A Gaussian representation of the electron density profile was used [43] with the following parameters being adjusted: z_H_ and σ_H_, the center and width of the head group Gaussian, respectively; σ_C_, the width of the hydrocarbon chain Gaussian; and ρ_R_, the ratio of the methyl-terminus electron density amplitude (ρ_C_) to the headgroup amplitude (ρ_H_), ρ_R_ = ρ_C_/ρ_H_. The headgroup-to-headgroup distance (d_HH_) is defined as 2 z_H_.

#### 2.2.7. Powder X-ray Diffraction (PXRD)

To analyze the structure and morphology of SeNPs incorporated into liposomes, PXRD measurements were performed. The PXRD patterns were recorded by an Aeris Benchtop X-ray diffractometer (Malvern Panalytical Ltd., Malvern, UK) with Ni-filtered copper radiation in Bragg–Brentano geometry. A drop of the sample was placed on a silicon zero-background holder and patterns were recorded in the range 2θ = 5–70° with a step size of 0.005° and 10 s per step. For data treatment, the PANalytical High Score Plus 4.5 software suite was used.

#### 2.2.8. Determination of Total Selenium

Microwave-assisted acid digestion of Lip-SeNPs was performed in an UltraClave IV High Performance Microwave Reactor (MLS GmbH, Leutkirch, Germany). Quantitative determination of Se was performed with an Agilent 7900 ICP-MS (Agilent Technologies, Waldbronn, Germany) equipped with a MicroMist nebulizer and a Scott type spray chamber. Sample solutions were introduced into the ICP-MS with an ASX-500 autosampler (Agilent Technologies). For digestion, 0.1 or 0.5 mL of the samples were transferred into the quartz vessels of the microwave digestion system. After addition of 2 mL HNO_3_ and 2 mL water the samples were digested at 250 °C for 30 min. The digests were then quantitatively transferred into 15 mL polyethylene tubes and filled up to 10 mL with water. All samples were digested in duplicate. Depending on the expected Se concentration, Se digests were either measured directly or after 1 + 9 or 1 + 99 dilution, respectively. Prior to analysis, an internal standard solution containing Ge and In in 2% (*v*/*v*) HNO_3_ was added to all samples (final concentration 10 µg/L). The ICP-MS was operated in the hydrogen reaction mode (3.5 mL H_2_/min). For enhancement of the Se signal, 1% CO_2_ in argon was used as optional gas at a flow rate of 12% of the carrier gas flow. Se was monitored on *m*/*z* 77, 78, 80 and 82 using an integration time of 0.3 s. Quantification was performed by external calibration with acid matched standard solutions using *m*/*z* 78 and ^72^Ge as internal standard. Quality control of total selenium determination was performed using a plasma sample from a round-robin exercise (reference value, 237.8 mg Se/L; range of tolerance, 206.3–269.3 mg Se/L, found value: 242 ± 11 mg Se/L, *n* = 6).

#### 2.2.9. Encapsulation Efficiency and Loading Capacity

The encapsulation efficiency (%EE) and the loading capacity (%LC) of Se into Lip-SeNPs were evaluated by measuring the amount of Se in Lip-SeNPs by ICP-MS in relation to the concentration of Se used for the synthesis of SeNPs loaded liposomes. % EE and % LC were calculated according to Equations (1) and (2), respectively:% EE = *γ* (Se, incorporated)/*γ*(Se, total) × 100(1)
% LC = *γ* (Se, incorporated)/*γ*(lipid) × 100(2)
where *γ* (Se, incorporated) represents the concentration of Se incorporated in Lip-SeNPs, *γ* (Se, total) is the total concentration of Se (selenium contained in Na_2_SeO_3_) used for the synthesis and *γ* (lipid) is the concentration of the lipids used for the assembly of SeNPs loaded liposomes.

#### 2.2.10. DPPH Assay

The radical scavenging capacity of Lip-SeNPs was tested by a DPPH assay according to Bai et al., in a slightly modified version using the stable free radical 2,2-diphenyl-1-picrylhydrazyl (DPPH) [44]. Briefly, 100 µL of Lip-SeNPs in PBS buffer (pH:7.4) were applied on a 96-well plate mixed with 100 µL of 0.4 mM ethanolic DPPH solution and incubated for 30 min in the dark at room temperature. The final concentrations were 25, 50 and 75 µg/mL SeNPs. A blank sample was prepared by replacing the DPPH solution with ethanol. Ascorbic acid (1 mM and 10 mM in PBS buffer, pH 7.4) was used as positive control while 100 µL DPPH diluted with 100 µL PBS was used as reference. The absorbance at 517 nm was measured using the Clariostar^®^ plate reader (BMG LABTECH GmbH, Ortenberg, Germany). The scavenging ability (•DPPH_scav_ %) was calculated according to the following equation:•DPPH_scav_ = (A_0_ − A_sample_ + A_blank_)/A_0_ × 100(3)
where A_0_ is the initial absorbance value of the free radical DPPH without Lip-SeNPs, A_sample_ is the absorbance of the DPPH radical after incubation with Lip-SeNPs and A_blank_ is the absorbance value of Lip-SeNPs without DPPH at 517 nm.

#### 2.2.11. Dissolution and Release Kinetics in Simulated Body Fluids

The dissolution and release kinetics of Se from Lip-SeNPs were measured after incubation with either simulated gastric fluid (SGF; containing 2 g sodium chloride, 3.2 g pepsin, 7 mL hydrochloric acid per litre; pH 1.2) or simulated intestinal fluid (SIF; containing 6.8 g potassium phosphate, 10 g pancreatin, 77 mL 0.2 M sodium hydroxide per litre; pH 6.8) prepared according to the U.S. Pharmacopeia. Briefly, Lip-SeNPs (3 mg/mL lipid) were diluted 1:1 (*v*/*v*) with SGF or SIF and incubated at 37 °C under shaking for certain time periods. After 0, 1, 2, 4 and 24 h, 450 µL of samples were withdrawn and diluted with PBS to 2 mL. Subsequently, the samples were centrifuged at 10,000× *g* for 5 min to precipitate Lip-SeNPs. Then, the supernatant was filtered at 4000× *g* for 10 min using Vivaspin 6 filter units (Satorius, Göttingen, Germany) with a cut-off of 100 kD, in order to investigate whether the SeNPs are dissolved after being released. The pore size of the ultrafiltration membrane is too small for released SeNPs to get across until the SeNPs dissolve to Se ions. Defined volumes of the filtrate containing dissolved Se ions were taken for quantification by ICP-MS. To determine the total amount of Se encapsulated in Lip-SeNPs, 10 µL of 10% Triton-X were added after 24 h of incubation to solubilize the membrane and to release the Se.

#### 2.2.12. In Vitro Adhesion and Uptake Studies

For the in vitro uptake studies a co-culture of Caco-2 and HT29MTX cell lines in the ratio of 7:3 was seeded on 12-well transwell inserts (Corning Costar polyester filters, pore size of 3.0 mm; Szabo Scandic, Vienna, Austria) with a seeding density of 5 × 10^5^ cells/well and cultivated for three weeks. Transepithelial electrical resistance (TEER) measurements were performed to check the integrity of the cell layer [43]. After washing with PBS, the cells were incubated with Lip-SeNPs diluted in PBS (final Se concentration of 20 μg/mL) at 37 °C for 4 h (*n* = 3). PBS buffer was used as blank. After incubation, the dispersions were removed and cells were washed twice with PBS. The cytoskeleton of the cells was stained with Alexa Fluor 488 Phalloidin (Thermo Fisher Scientific, Waltham, MA, USA) and the nuclei were counterstained with Hoechst 33,342 (Thermo Fisher Scientific). The membrane of the transwells was cut out, put on a glass slide, fixated with mounting media and covered with a cover glass. Images were acquired with a LSM 510 Meta confocal laser scanning microscope (Carl Zeiss GmbH, Vienna, Austria) with a Zen2008 software package. Alexa Fluor 488 Phalloidin dyed cytoskeleton was detected at 488 nm excitation wavelength using a BP 505–550 nm bandpass detection for the green channel. Hoechst 33,342 stained nuclei were visualized at an excitation wavelength of 405 nm using a BP 420–480 nm bandpass detection for the blue channel. The fluorescent rhodamine phospholipid labelled Lip-SeNPs were detected at 543 nm excitation wavelength using a long pass (LP) detection for the red spectral region. Images of randomly chosen areas of the cell monolayers were captured via CLSM and Z-scans were acquired for each µm in depth. The thickness of the cell layer was between 20 and 28 µm.

To evaluate the permeability of Se through the intestinal co-culture model, a 12-well transwell system was used as described above. The cells were incubated with Lip-SeNPs (final Se concentration of 20 μg/mL) for 4 h before 400 µL and 1000 µL of samples were taken from the donor and the acceptor compartment, respectively. To determine the cellular uptake, the cells were washed with PBS and lysed with a 2% Triton-X solution. The Se concentration was determined with ICP-MS as described above (*n* = 2).

#### 2.2.13. In Vitro Cytotoxicity Studies

To determine potential cytotoxic effects of Lip-SeNPs, the cell membrane integrity was measured by determining the lactate dehydrogenase (LDH) release. For these measurements, the co-cultured cells were seeded (6 × 10^4^ cells/well) in 96-well plates (Greiner Bio-One GmbH, Frickenhausen, Germany). After 24 h, the medium was replaced by the samples dispersed in serum-free medium and incubated for 4 h. Subsequently, a CytoTox-ONE Homogeneous Membrane Integrity Assay (Promega^®^ Corp., Madison, WI, USA, 53,711) was used according to the manufacturer’s instructions and the fluorescence was measured at an excitation wavelength of 560 nm and an emission wavelength of 590 nm using a UV-/VIS plate reader (Fluostar Galaxy, BMG Labtech, Ortenberg, Germany). Control wells for 100% LDH release were treated with 2 µL of a lysis solution and all results were blank corrected. The data were evaluated as described previously [45].

#### 2.2.14. Ex Vivo Mucoadhesion and Uptake Studies

Ex vivo mucoadhesion and uptake studies were performed with freshly excised porcine small intestine (purchased from Marcher Fleischwerke, Graz, Austria). The tissues were transported to the laboratory in Krebs buffer at a temperature of 4 °C. Porcine intestinal tissues were rinsed with PBS and cut longitudinally into planar sheets and mounted in static Franz diffusion cells (PermeGear, Hellertown, PA, USA) between the receptor and donor compartments. The epithelium faced the donor compartment (apical mucosal side). To obtain a detectable fluorescence signal, 1 mL Lip-SeNPs with a lipid concentration of 1.3 mg/mL was applied and incubated for 4 h (*n* = 2). Afterwards, the mucosa was washed three times with PBS and fixed with formalin solution (neutral buffered, 10% (*v*/*v*)). The samples were shock frozen in Tissue-Tek1 O.C.T. Compound (Sakura Finetek USA Inc, St. Torrance, CA, USA) and cut into 10 µm slices with a cryo-microtom (Microm HM 560, Thermo Fisher Scientific). Images were acquired via fluorescence microscopy (Olympus BX-51, camera: DP-71; Olympus Austria Ges.m.b.H, Vienna, Austria) with an excitation wavelength of BP 520–550 nm and an emission wavelength of LP 580 nm for red fluorescence.

#### 2.2.15. Statistical Analysis

All values are expressed as means ± standard deviation (SD). The number of repetitions performed for each experiment is indicated. The differences between groups were analyzed by an unpaired Student’s *t*-test using the program SPSS Statistics (IBM Corp. Version 27.0. 1.0, Armonk, NY, USA). A value *p* ≤ 0.05 was considered as statistically significant with *, **, *** indicating *p* < 0.05, *p* < 0.01, *p* < 0.001, respectively.

## 3. Results and Discussion

Oral administration of drugs is one of the most exploited and preferred administration routes. However, harsh conditions in the GI microenvironment and insufficient bioavailability of the administered drugs might hamper their oral applicability. The use of biocompatible delivery vehicles may impede substance degradation and inactivation in the GI tract. Thus, the main motivation for this study was to design a hybrid thiolated chitosan coated liposomal delivery system with mucoadhesive properties encapsulating SeNPs. The intention was to improve the biological profile of SeNPs for proper uptake of Se in the microenvironment of the small intestine. Thereby, Se should exert beneficial effects in the GI tract in support of common therapies.

### 3.1. Synthesis and Physicochemical Characterisation of Lip-SeNPs

As a starting point for synthesis, a highly promising mucoadhesive thiochitosan based delivery system described by Gradauer et al. [28] was refined and adapted. Cystein modified chitosan was synthesized as described previously [37]. In total, 233 ± 11 µmol thiol groups were covalently attached per gram of polymer. The Cs-NAC was further characterized by FT-IR and ^1^H NMR. After modification, the absorption band in the FT-IR spectrum of chitosan was shifted from 1660 to 1625 cm^−1^. The appearance of a new amide peak at 1520 cm^−1^ (bending vibration of C-N-H, stretching vibration of C-N) and the shoulder in the range of 2590 and 2480 cm^−1^, belonging to the thiol group in NAC, clearly confirmed the success of modification (Appendix A). In the ^1^H NMR spectrum of the Cs-NAC, a chemical shift was observed at 2.94 ppm belonging to the methylene protons next to the thiol moiety, while the one at 4.47 ppm corresponds to the methine protons in NAC. At 2.02 ppm, the peak of the acetyl protons appeared with high intensity, as an indication of the excess of acetyl groups, compared to native chitosan (Appendix A). In this study, a new manufacturing protocol was established by using a microfluidics rapid mixing technology. This technique enabled the chemical reduction of selenite (Na_2_SeO_3_) to elementary Se, the formation of SeNPs, the assembling of liposomes and the incorporation of SeNPs into liposomes simultaneously by a single step procedure. The reaction of SeNPs formation is presented by Equation (4):SeO_3_^2−^ + 2C_6_H_8_O_6_ + 2 H^+^ → Se + 2C_6_H_6_O_6_ + 3H_2_O(4)

Immediately after the assembling process of Lip-SeNPs, the color of the solution turned orange, which was taken as indication for successful reduction of selenite to elemental Se. The remaining Se, which stayed in its ionic form (Na_2_SeO_3_), was removed by dialysis, which was performed immediately after particle synthesis and before covalent coupling of Cs-NAC (see Figure 2). Non-encapsulated SeNPs were unstable in aqueous solution and precipitated before dialysis. By this synthesis procedure an encapsulation efficiency (EE%) of 37.3 ± 5.8 wt% Se and a loading capacity (LC%) of 10.6 ± 1.5 wt% Se (*n* = 4) were achieved using an initial concentration of 13 mM Se. EE% of about 30 wt% were comparable to EE% reported by Gradauer et al. [31], who has used FITC calcitonin as a model drug. Here it is important to note that throughout the manuscript 1 mg/mL Lip-SeNPs corresponds to a concentration of 40 µg/mL encapsulated SeNPs.

One of the greatest advantages of microfluidics-based synthesis of liposomes is the simple preparation procedure. There is no need for size adjustment by membrane extrusion, as normally required for conventional procedures such as dry lipid film rehydration techniques [44,45]. Selecting appropriate flow rates and flow ratios enables the preparation of unilamellar liposomes of a defined size [44]. With the chosen flow rate ratio between aqueous and organic phase and a total flow rate of 5 mL/min Lip-SeNPs with a mean diameter of 72 ± 36 nm (*n* = 3) and 90% of the particles (D90) being smaller than 91 ± 47 nm were obtained. Upon polymer conjugation through the covalent binding reaction of the free –SH groups of the polymer to the maleimide functionalized lipid molecules (DOPE-MCC) the particle size increased significantly (*p* < 0.05). The thiolated Lip-SeNPs were more heterogeneous with an average size of 253 ± 134 nm (*n* = 3). About 90% of the thiolated Lip-SeNPs were in a size range below 436 ± 253 nm, as expressed by the D90 value. The data were in good agreement with size distributions reported previously for the chemical coupling of liposomes to thiolated chitosan [31]. Similar results were also obtained in the study of Li et al. [46] in which thiolated chitosan coated liposomes were used as delivery vehicle for curcumin. In general, the size of the designed hybrid thiolated liposomal delivery system for SeNPs is within an acceptable range reported for intestinal absorption [47,48].

Aggregation behavior of thiolated Lip-SeNPs was followed with time in PBS over 14 days upon storage at 4 °C. Thiolated Lip-SeNPs can be regarded as colloidally stable for at least two weeks as no precipitate was observed, and the particles displayed only slight but not significant changes (*p* > 0.1) in size (Appendix A).

The size values derived from NTA measurements were compared to morphological data obtained by TEM (Figure 3). Uncoated Lip-SeNPs displayed a spherical shape, typical for plain liposomes. The darker spots seen in the middle of the liposomes could represent the SeNP-filled core of the liposomes having a higher electron density than the surrounding lipid shell (Figure 3a). The particle sizes seemed to be smaller than determined by NTA, which could be explained by some shrinking during sample preparation including staining and drying procedures. The thiolated Lip-SeNPs (Figure 3b) were larger due to the polymer coat, which was visible as irregular structure on the particle surface.

Zeta potential values may provide information on the surface charge and stability of dispersed particles in solution, but they depend on the ionic strength and the pH value of the medium [49]. Measured at standardized conditions, the uncoated Lip-SeNPs had a negative zeta potential of −11.6 ± 2.2 mV (*n* = 3), while thiolated Lip-SeNPs showed a positive zeta potential of 12.3 ± 1.0 mV (*n* = 3), which is due to the remaining free functional -SH groups of the polymer after coupling. The initial amounts of free -SH groups in Cs-NAC was determined after incubation of the polymer with POPC liposomes lacking maleimide groups for covalent coupling. For the Cs-NAC polymer, an amount of 233 ± 11 nmol free -SH groups per mg polymer (*n* = 3) was detected, which corresponds to 320 ± 27 nmol free -SH groups per mg lipid (*n* = 6). In the thiochitosan conjugated samples, an amount of 236 ± 27 nmol free -SH groups per mg lipid (*n* = 6) was found. The mean difference of 84 nmol free -SH groups/mg lipid (*p* = 0.00024) found after the coupling reaction clearly indicated that the covalent binding of the polymer onto the surface of the liposomes was successful, but a considerable amount of free -SH groups is still available. The remaining free -SH groups are important for the adhesion of the particles to the mucus layer, where the free -SH groups of the polymers are supposed to interact with the cysteine residues of mucin glycoproteins via disulfide bridge formation [50].

To obtain information on the lipid bilayer organization of Lip-SeNPs and to determine whether polymer coating has affected the bilayer integrity, Synchrotron SAXS experiments were performed on uncoated and thiolated Lip-SeNPs. Both scattering curves (Figure 4) were fitted to a unilamellar vesicle model by using a single bilayer electron density model. The electron density profile is shown (Figure 4, inset) and resultant fitting parameters are summarized in Appendix A. The thiolated Lip-SeNPs showed a slightly enhanced phospholipid headgroup-to-headgroup distance (dHH) as compared to uncoated Lip-SeNPs. In addition, the first order maximum for the thiolated Lip-SeNPs was shifted to lower q-values indicating larger structures. However, both scattering curves were indicative for unilamellar or uncorrelated bilayer structures, meaning that the polymer coating had low impact on the membrane organization of the particles.

The composition and phase behavior of the SeNPs inside Lip-SeNPs were determined by PXRD measurements yielding diffraction patterns consistent with those reported in literature for chitosan embedded SeNPs [51]. The broad diffraction peaks at 2 θ/° at 28 and 41 correspond to (101) and (110) lattice planes, respectively, indicating the presence of both crystalline and amorphous phases of Se embedded within Lip-SeNPs (Figure 5).

### 3.2. Radical Scavenging Capacity

To assess the free radical scavenging (FRS) ability of Lip-SeNPs, the particles were incubated with the stable free radical DPPH. In general, the radical scavenging activity of antioxidants can be evaluated from the decrease in absorbance of the DPPH radical when the stable nitrogen radical becomes inactivated by the exchange of a hydrogen to produce the corresponding hydrazine DPPH_2_ [52]. In our experimental setup, the absorbance decreased in a concentration dependent manner by increasing the Lip-SeNPs concentration from 25 to 50 and 75 µg/mL SeNPs (Figure 6). For uncoated LipSeNPs the scavenging activity increased from 27.1 ± 6.2 to 37.7 ± 0.2%, while the scavenging potential for thiolated Lip-SeNPs was significantly higher, increasing from 42.0 ± 8.2% to 68.5 ± 7.2% with increasing concentrations of SeNPs from 25 to 75 µg/mL selenium. For comparison, ascorbic acid used as the positive control with concentrations of 1 mM and 10 mM, scavenged 44.4 ± 0.3 and 94.4 ± 0.8% of DPPH, respectively. The relatively high FRS potential of thiolated Lip-SeNPs could be partially explained by the anti-oxidative and free radical scavenging potential of chitosan as shown in previous studies [53,54,55]. The proposed mechanisms for the FRS potential of chitosan relies on hydrogen abstraction, addition reactions and electron transfer, whereby the structure and molecular weight of Cs as well as the nature of free radicals dictate the FRS mechanism [55]. Similarly, NAC as the cysteine prodrug and glutathione (GSH) precursor has been recognized as the biologically active compound that displays anti-oxidative properties and exhibits modulatory effects on oxidative stress- and inflammation-based diseases [56]. The suggested NAC’s mode of action is through interaction of free thiol groups with free radicals or as source for GSH biosynthesis. Along these lines, thiolated Lip-SeNPs demonstrated advanced FRS properties that can be considered for the design of complex co-administration vehicles entrapping anti-oxidative and anti-inflammatory drugs combined with SeNPs, to achieve further synergistic protective effects.

### 3.3. Release Kinetics of Selenium in Simulated Body Fluids

Even though the thiolated Lip-SeNPs are designed to release Se in a sustained manner once bound to the mucus, the Lip-SeNPs have to be stable in the gastric environment to prevent the aggregation of hydrophobic SeNPs. Moreover, interaction with enzymes present in body fluids could affect the surface properties of Lip-SeNPs. Thus, the release kinetics of Se and the stability of thiolated Lip-SeNPs in simulated body fluids mimicking the gastrointestinal environment was measured upon incubation with SGF and SIF over a period of 24 h. The data showed that the release rate of Se was very low and most of the SeNPs were retained within the liposomes independent of the media. For all tested systems, the time-dependent leakage of Se over a period of 24 h was less than 5% (Appendix A). Although the release rate in SIF was slightly faster than in SGF (*p* > 0.1), the release kinetics of Se was found to be rather insensitive to changes in the pH value (pH: 1.2 and 6.8) of the media. Moreover, Lip-SeNPs were not prone to enzymatic degradation to permit an effective protection of SeNPs in the microenvironment of the GI tract.

### 3.4. In Vitro and Ex Vivo Particle Uptake Studies

To investigate the cellular interaction and uptake behavior of Lip-SeNPs, two different models were used—an in vitro cell co-culture model and an ex vivo model of porcine small intestine. For both approaches, the particles were labelled with a fluorescent rhodamine phospholipid dye, which was incorporated into the lipid bilayer. The CLSM images showed the interaction of Lip-SeNPs with the co-culture cell system (Caco-2:HT29MTX in a ratio of 7:3) after incubation for 4 h at 37 °C. The images revealed that the thiolated Lip-SeNPs were predominantly located in between the cells (Figure 7a, top) whereas the uncoated Lip-SeNPs seemed to be taken up into the cells (Figure 7b). This was also visible in the Z-stacks, in which thiolated Lip-SeNPs were located on top of the cell layer and did not deeply penetrate into the intestinal mucus (Figure 7a, bottom) in contrast to uncoated Lip-SeNPs, which were used as control (Figure 7b, bottom).

To exclude possible cytotoxic effects of the mucoadhesive thiolated Lip-SeNPs, in case the particles are taken up by enterocytes, an LDH assay was performed. Although it is rather unlikely that the relatively large chitosan coated Lip-SeNPs may be internalized as a whole, we wanted to address the question of whether SeNPs, which are dissolved within 4 h of incubation to release free Se, are cytotoxic to cells. Again, the Caco-2:HT29-MTX co-culture model was used to mimic the intestinal epithelial barrier and SeNP loaded thiolated Lip-SeNPs were compared to empty liposomes. We found that none of the samples showed negative effects on the cell membrane integrity in the applied concentration range up to 1 mg/mL thiolated Lip-SeNPs, corresponding to a Se concentration of about 40 µg/mL (Appendix A). After an incubation time of 4 h, all samples showed an LDH release of less than 5% compared to the cell lysis, which was taken as 100% release. The rather low cytotoxicity observed for SeNPs encapsulated into liposomes as compared to results reported in literature [57] for even lower concentrations of Se might be explained by the protective effects of the phospholipid bilayer rather than the chitosan coat. Thus, apart from the Se concentration and exposure time to cells, it can be assumed that toxicities reported for SeNPs strongly depend on particle stabilization, surface coatings and finally on the dissolution rate and release of free Se ions [21].

To address the issue of Se release and transport properties crossing the intestinal barrier model described above, the particles were incubated with the cells on transwells for 4 h. The Se concentration in the apical and basolateral compartment, as well as in the cell lysate (*n* = 2; run in duplicate) showed that approximately 80% of Se could still be found in the apical compartment after an incubation time of 4 h. About 14% of Se was found in the basolateral compartment and about 6% in the cell lysate. This behavior is in line with data obtained from the dissolution experiments, which showed that only about 0.8% of Se was released in SIF after 4 h incubation (see Appendix A). All these data suggest that the release and transfer of Se across the intestinal barrier is quite slow. We assume that the positively charged thiolated Lip-SeNPs adhered to the cell surface and remained embedded in the cell membranes, as evidenced by CLSM, without notable release of their payload as detected by the retained Se in the cellular microenvironment. Thereby, the free thiol-groups of thiolated chitosan are suggested to interact with the cysteine substructures of mucin proteins in mucus thus forming disulfide bridges [23]. Once the thiolated Lip-SeNPs are in contact with the mucus the particles are trapped in the mucus gel layer. Considering the rapid turnover time of the intestinal mucus, which was estimated to be less than 270 min [58], the high retention rate observed in our study could lead to a clearance of the Lip-SeNPs from the targeted site in the intestinal tract before Se is quantitatively released. Thus, it would be important to establish proper conditions at which mucoadhesive and lipid-membrane destabilization effects are balanced to enable SeNPs dissolution and release in the GI tract. Therefore, an efficient permeation of selenium ions through the intestinal barrier should be guaranteed. One option to make the lipid bilayer more unstable in the presence of pancreatic enzymes or bile acids present in the GI tract is that small amounts of stearylamine could be added to the LipSeNP formulation [59]. Stearylamine has a positive charge and tends to interact with negatively charged cholate molecules to break up the tight lipid bilayer organization and subsequently to promote drug release. The drugs, in our case dissolved Se ions, are released into the mucus layer upon degradation of the liposome and pass through the intestinal epithelial cell layer by passive diffusion [60]. The results of the in vitro adhesion study are comparable with data obtained in ex vivo studies. Figure 8 shows the fluorescence images of the ex vivo experiments on histological sections of porcine small intestinal mucosa. For thiolated Lip-SeNPs most of the red signal is located on the top layer of the mucosa in accordance with literature data [61] showing that thiolated chitosan becomes immobilized on the mucus layer due to the formation of disulfide bridges between the free SH groups of the Cs-NAC and the cysteine residues of the mucin glycoproteins located on the surface of the mucus layer. Since the fluorescent signal originated from the liposomes it can be assumed that the Lip-SeNPs assembly remains intact upon interaction with the mucus. In contrast, uncoated Lip-SeNPs used as control (Figure 8b) did not adhere to the mucus layer and were washed away. The results of both experiments, in vitro and ex vivo, revealed that thiolated Lip-SeNPs were mainly attached to the mucus layer. This fits very well with data reported in literature, showing that thiolated chitosan exhibits enhanced mucoadhesive properties [25,62].

## 4. Conclusions

SeNPs were successfully synthesized and loaded into liposomes in a one-step microfluidics manufacturing process. The incorporation of SeNPs into liposomes is intended to facilitate the transport of hydrophobic SeNPs in an aqueous environment, simultaneously preventing alterations of SeNP size, charge or rapid ion release that would affect the bioavailability of SeNPs. The liposomes were coated with a thiolated chitosan to increase stability and retention of Lip-SeNPs in the gastrointestinal microenvironment. The free -SH groups of the thiolated chitosan are suggested to bind to the exposed cysteine residues of the mucus glycoproteins to promote mucoadhesion. At least in the simulated GI-model used in this study, the dissolution and release of selenium from the Lip-SeNPs was too slow to enable an efficient transfer of selenium across the intestinal epithelium. Nevertheless, our findings suggest that mucoadhesive thiolated liposomes could serve as a refined drug delivery system for SeNPs to improve the oral bioavailability of selenium, but the liposomes need to be degraded once adsorbed to the mucus layer. To exploit the full therapeutic potential of thiolated Lip-SeNP formulations, however, a co-administration with anti-inflammatory or anti-cancer drugs incorporated within the same liposome particle should be considered for the future, with the goal to stimulate a better immune response upon synergistical action of Se with a therapeutic agent.

## Figures and Tables

**Figure 1 pharmaceutics-14-00803-f001:**
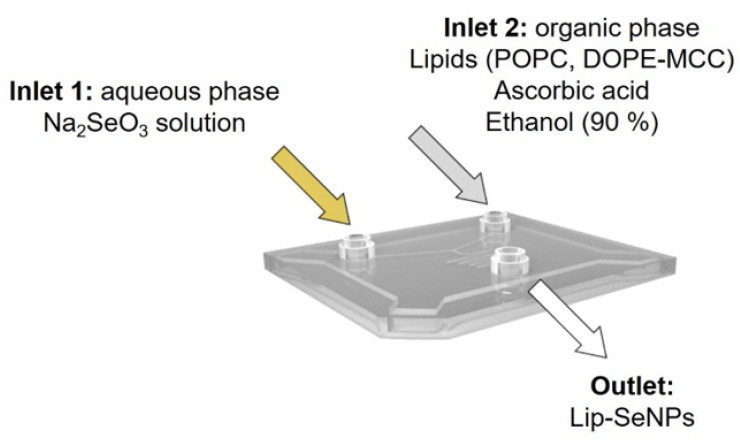
Scheme of the microfluidic cartridge used for the simultaneous chemical reduction and assembling process of SeNPs loaded liposomes. The components for the synthesis were injected into the inlets as shown above; through the outlet, Lip-SeNPs were obtained. A flow ratio of 2.5:1 aqueous phase to organic phase and a flow rate of 5 mL/min were used.

**Figure 2 pharmaceutics-14-00803-f002:**
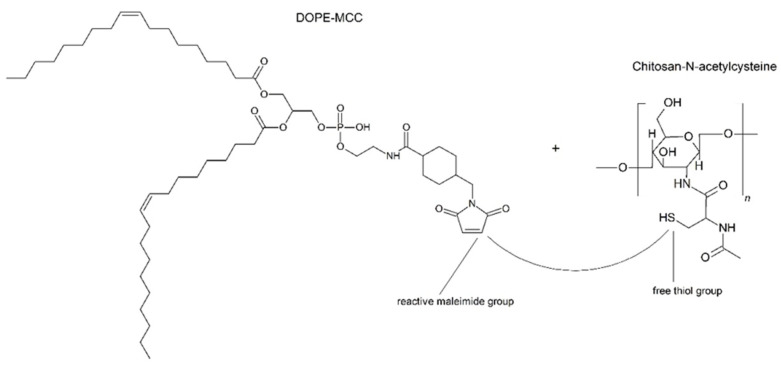
Reaction scheme for thiomer coupling. The free thiol groups of chitosan-N-acetylcysteine (Cs-NAC) are covalently bound to the maleimide groups of DOPE-MCC.

**Figure 3 pharmaceutics-14-00803-f003:**
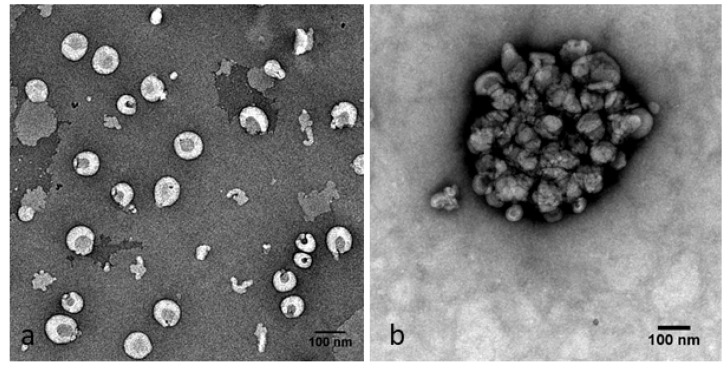
TEM images of (**a**) uncoated and (**b**) thiolated Lip-SeNPs. The latter show an irregular structure of the polymers on the surface of the Lip-SeNPs. The scale bars are 100 nm.

**Figure 4 pharmaceutics-14-00803-f004:**
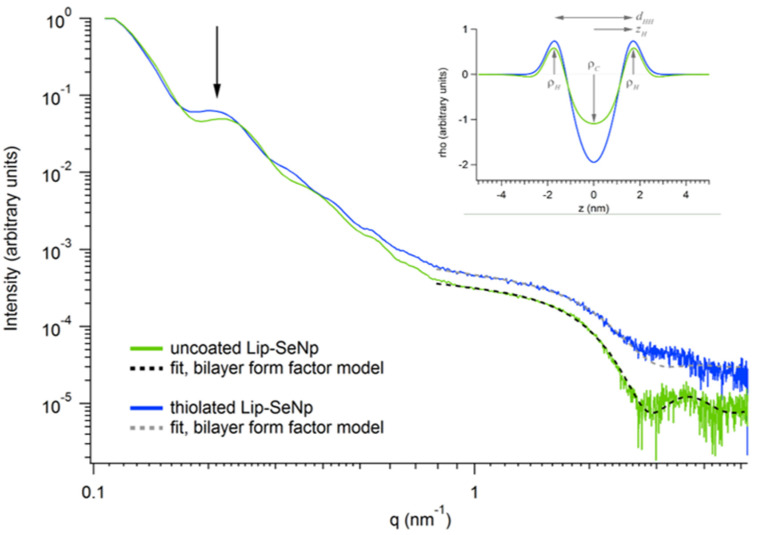
Scattering curves and electron density distribution (inset) of uncoated (green curve) and thiolated (blue curve) Lip-SeNPs fitted with a bilayer form factor model. The arrow indicates the shift of the first scattering maximum of the thiolated Lip-SeNPs to lower q-values. The electron density distribution (inset) shows a very slight change in the bilayer thickness.

**Figure 5 pharmaceutics-14-00803-f005:**
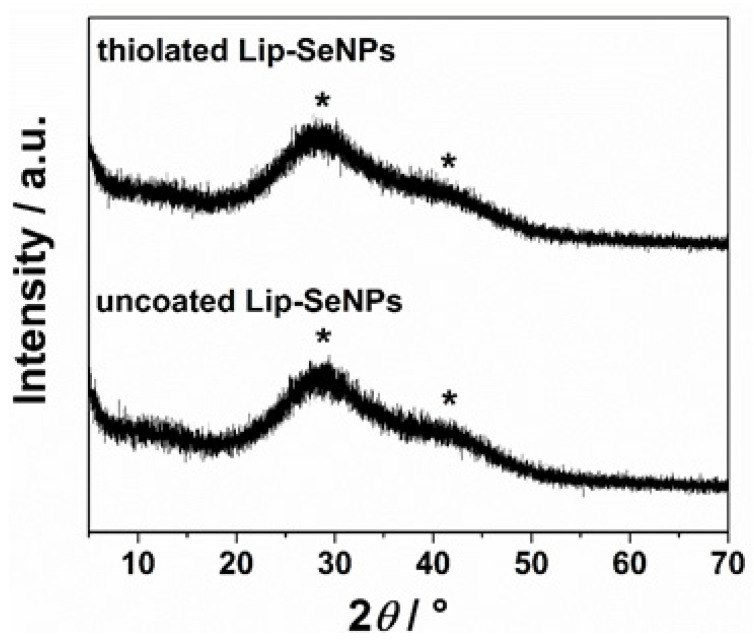
PXRD diffraction pattern of Lip-SeNPs. The stars (*) indicate the positions of the diffraction peaks corresponding to (101) and (110) lattice planes, respectively.

**Figure 6 pharmaceutics-14-00803-f006:**
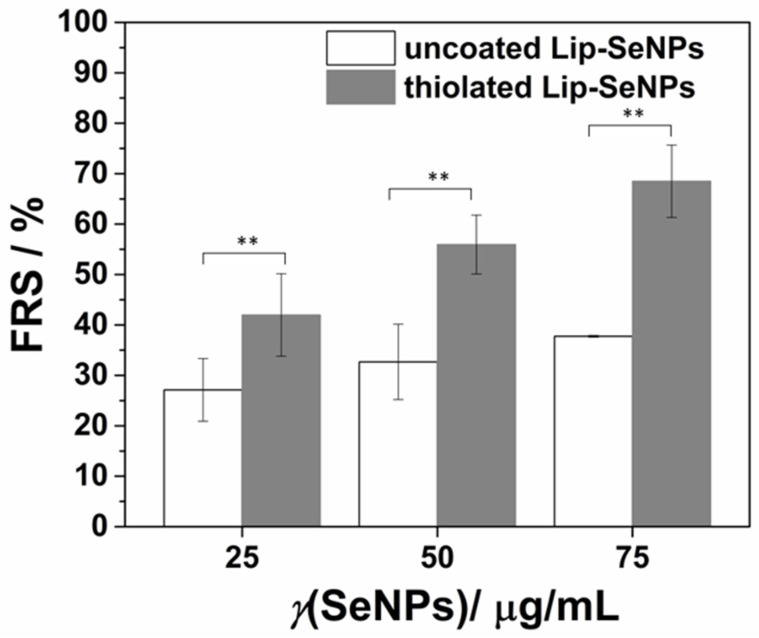
Free radical scavenging (FRS) ability of Lip-SeNPs. The values are expressed as mean values ± SD (*n* = 3); unpaired *t*-test, ** *p* < 0.01.

**Figure 7 pharmaceutics-14-00803-f007:**
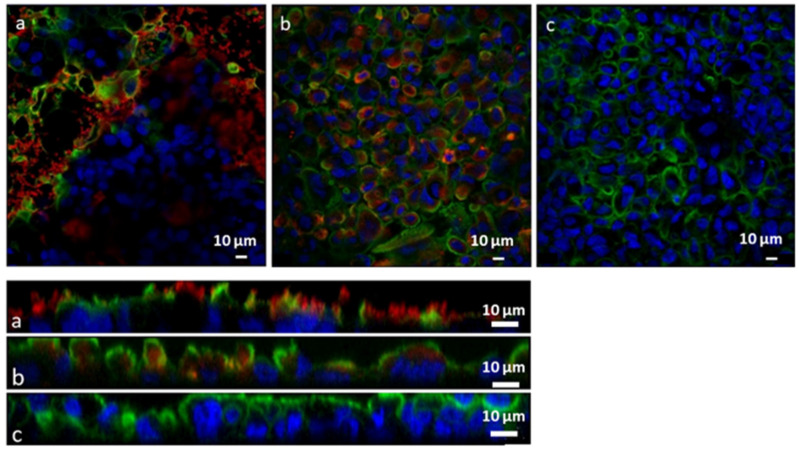
CLSM images (**top**, **a**–**c**) and Z-scans (**bottom**, **a**–**c**) of the cell co-culture (Caco-2:HT29MTX; 7:3) treated with 500 µg/mL particle suspension for 4 h at 37 °C. (**a**) Thiolated Lip-SeNPs; (**b**) uncoated Lip-SeNPs; (**c**) cell control. The particles (phospholipid layer) were labelled with rhodamine B (red), the cell nuclei were stained with Hoechst (blue) and the cell membranes were stained with Alexa Fluor 488 Phalloidin (green). The scale bars are 10 µm.

**Figure 8 pharmaceutics-14-00803-f008:**
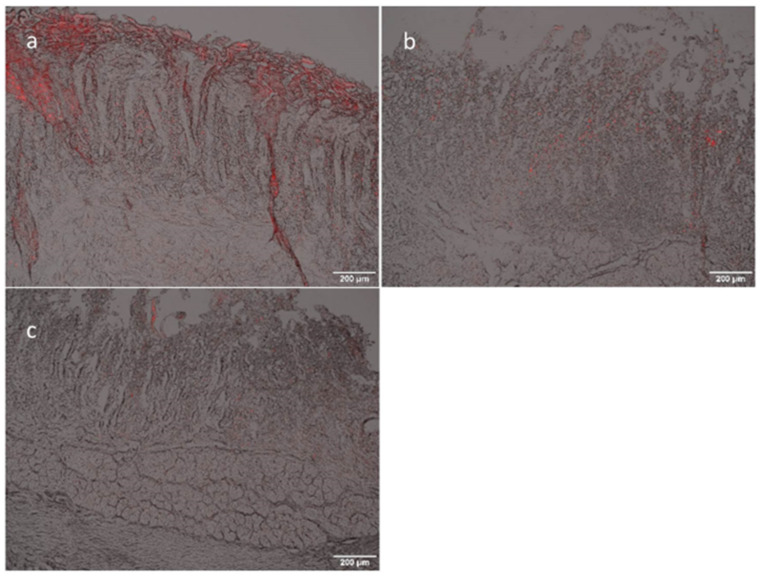
Fluorescence microscopy micrographs of a histological section of porcine small intestinal mucosa; the red fluorescence signal shows rhodamine B labelled particles; (**a**) thiolated Lip-SeNPs; (**b**) uncoated Lip-SeNPs; (**c**) blank. The reddish background color comes from the auto-fluorescence of the mucosa; scale bar: 200 µm.

## Data Availability

Not applicable.

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
