# Peer review of "Thiolated Chitosan Conjugated Liposomes for Oral Delivery of Selenium Nanoparticles"

_pharmaceutics, 2022, doi:10.3390/pharmaceutics14040803_

Round 1

Reviewer 1 Report

This work turns out to be quite complete. There are minor errors such as

-In equation 1 it should be % EE ..¿?

-In lines 514 to 517 there are different font sizes.

-In figure 3, there are things that are not clear. In the 3c and d image, hardly anything can be seen, if nothing is observed, it would be better not to show them. 

Author Response

We thank the reviewer for carefully reading our manuscript.  We have corrected the minor errors.
Figure 3: Due to the poor quality of the SEM images, which has also been criticized by other reviewers, we have decided not to show the SEM data as we believe that no information will be lost.

Reviewer 2 Report

Dear Researcher, please revise your manuscript as following suggested points

I strongly recommend revising this manuscript as follows:

  1. The author should add the information about the in thiolated chitosan what was the degree of substitution utilized.
  2. The author should add the information about NP polymer author can refer to this article and can be cited in the introduction. https://doi.org/10.1002/chem.201701900, https://doi.org/10.1021/acs.biomac.0c00889
  3. Figure no 3 . TEM images of (a) uncoated and (b) thiolated Lip-SeNPs. The latter show an irreg- ular structure of the polymers on the surface of the Lip-SeNPs. SEM images of (c) uncoated Lip-SeNPs show the spherical topology of plain liposomes, while (d) thiolated Lip-SeNPs  seem to be embedded in a polymer cloud. The author should change with the more clear image.
  4. The NMR and FT-IR spectra of thiolated chitosan should add to the manuscript.

These all recommended points can be referred to from suggested research articles. 

Reviewer 3 Report

The article is well represented by the authors.

The aim of the work is well developed in the description of the results and the conclusion.

CLSM images 7 (a) and 7 (b) of the cell co-culture (Caco-2:HT29MTX) do not clearly show the Scale bar.

Author Response

We thank the reviewer for carefully reading our manuscript. The scale bars in Figure 7 are now clearly shown for all images.

Reviewer 4 Report

In the present MS, the thiolated chitosan conjugated liposomes for oral delivery of selenium nanoparticles was prepared and evaluated.

  1. The characteristic of Se NP should supplied in detail, such as the mechanism of forming Se NP which from Na2SeO3, the reacting material for chemical reduction of selenite (Na2SeO3) to elementary Se, the particle size and distribution of Se NPs.
  2. For uptake experiment, which was absorbed, the Se elementary, Se NP, or Se NP liposome?

Author Response

We thank the reviewer for carefully reading our manuscript.

In this study, SeNPs loaded liposomes were prepared in one step chemical reduction synthesis. The reaction of SeNPs formation is presented by equation (1)

SeO32- + 2C6H8O6 + 2 H+ → Se +2C6H6O6 + 3 H2O                                                   (1)

There are numerous studies in the literature that report the encapsulation of metallic NPs such as selenium, gold and silver NPs into liposomes [1-4]. The film hydrated method was used to prepare liposomes and subsequently, previously synthesized metallic NPs were introduced, followed by the lipid film rehydration. In the present study, the microfluidic approach, i.e. Nanoassemblr was utilized to prepare SeNPs loaded liposomes. The organic phase was composed of lipids and ascorbic acid dissolved in ethanol, while aqueous phase consisted Na2SeO3 dissolved in water. The molar ratio of ascorbic acid in organic phase and Na2SeO3 in aqueous phase was 2.5:1. Since the synthesis was performed in one step, the characterization was focused on SeNPs loaded liposomes.

References

  1. Qian Xie, Wenji Deng, Xue Yuan, Huan Wang, Zhiguo Ma, Baojian Wu, Xingwang Zhang, Selenium-functionalized liposomes for systemic delivery of doxorubicin with enhanced pharmacokinetics and anticancer effect. European Journal of Pharmaceutics and Biopharmaceutics, 122, 2018, 87. Doi: org/10.1016/j.ejpb.2017.10.010.
  2. Gabriel Charest, Tippayamontri Thititip, Shi Minghan, Mohamed Wehbe, Malathi Anantha, Marcel Bally, and Léon Sanche, Concomitant Chemoradiation Therapy with Gold Nanoparticles and Platinum Drugs Co-Encapsulated in Liposome. International Journal of Molecular Sciences 21, 2020, 4848. Doi: org/10.3390/ijms21144848.
  3. BartoszSkóra, Tomasz Piechowiak, Konrad A. Szychowski, JanGmiński, Entrapment of silver nanoparticles in L-α-phosphatidylcholine/cholesterol-based liposomes mitigates the oxidative stress in human keratinocyte (HaCaT) cells. European Journal of Pharmaceutics and Biopharmaceutics, 166, 2021, 163. Doi: org/10.1016/j.ejpb.2021.06.006.
  4. Zahraa S. Al-Ahmady, Roberto Donno, Arianna Gennari, Eric Prestat, Roberto Marotta, Aleksandr Mironov, Leon Newman, M. Jayne Lawrence, Nicola Tirelli, Marianne Ashford, Kostas Kostarelos, Enhanced Intraliposomal Metallic Nanoparticle Payload Capacity Using Microfluidic-Assisted Self-Assembly. Langmuir, 35(41),2019, 35, 41, 13318. Doi: org.10.1021/acs.langmuir.9b00579.

Line 457- 469: Immediately after the assembling process of Lip-SeNPs, the colour of the solution turned orange, which was taken as indication for successful reduction of selenite to elemental Se. The remaining Se, which stayed in its ionic form (Na2SeO3), was removed by dialysis, which was performed immediately after particle synthesis and before covalent coupling of Cs-NAC.  Non-encapsulated SeNPs were unstable in aqueous solution and precipitated before dialysis. 

We could not determine the particle size of the uncoated agglomerated SeNPs.

For the uptake studies, Cs-NAC coated liposomes encapsulating SeNPs were used and the dissolution and release kinetics of elementary selenium was recorded.

Round 2

Reviewer 4 Report

Accept